# Large Number of Direct or Pseudo-Direct Band Gap Semiconductors among *A*_3_*TrPn*_2_ Compounds with *A* = Li, Na, K, Rb, Cs; *Tr* = Al, Ga, In; *Pn* = P, As

**DOI:** 10.3390/molecules29174087

**Published:** 2024-08-28

**Authors:** Sabine Zeitz, Yulia Kuznetsova, Thomas F. Fässler

**Affiliations:** Chair of Inorganic Chemistry with Focus on Novel Materials, School of Natural Science, Technical University of Munich, Lichtenbergstraße 4, D-85747 Garching, Germany; sabine.zeitz@tum.de (S.Z.); yulia.kuznetsova@tum.de (Y.K.)

**Keywords:** Zintl phase, direct band gap, structure prediction, semiconductor, DFT calculation

## Abstract

Due to the high impact of semiconductors with respect to many applications for electronics and energy transformation, the search for new compounds and a deep understanding of the structure–property relationship in such materials has a high priority. Electron-precise Zintl compounds of the composition *A*_3_*TrPn*_2_ (*A* = Li − Cs, *Tr* = Al − In, *Pn* = P, As) have been reported for 22 possible element combinations and show a large variety of different crystal structures comprising zero-, one-, two- and three-dimensional polyanionic substructures. From Li to Cs, the compounds systematically lower the complexity of the anionic structure. For an insight into possible crystal–structure band–structure relations for all compounds (experimentally known or predicted), their band structures, density of states and crystal orbital Hamilton populations were calculated on a basis of DFT/PBE0 and SVP/TZVP basis sets. All but three (Na_3_AlP_2_, Na_3_GaP_2_ and Na_3_AlAs_2_) compounds show direct or pseudo-direct band gaps. Indirect band gaps seem to be linked to one specific structure type, but only for Al and Ga compounds. Arsenides show smaller band gaps than phosphides due to weaker *Tr*-As bonds. The bonding situation was confirmed by a Mullikan analysis, and most states close to the Fermi level were assigned to non-bonding orbitals.

## 1. Introduction

Semiconductors have a variety of applications, like solar cells, LEDs, laser applications, etc., which are essential for the continuous evolution of our modern society. As their applications become more diverse and complex, efficient band gap engineering is becoming increasingly important in the design of new and highly specialised materials [1,2,3]. When searching for compounds that allow a correlation between their atomic structure and the nature of their constituent elements with their properties, compound classes that have several or many representatives are particularly useful for determining the relevant parameters [4,5].

The Ga_x_Al_1−x_As and In_x_Ga_1−x_As compound family are exemplary compounds where a simple variety in the composition leads to highly tuneable band gaps [6,7,8,9,10]. However, besides these well-defined examples, concepts for more complex compound families are missing; thus, exploring possible crystal–structure band–structure relations is key for the future of intelligent and efficient material design. In a previous study, we reported that, for a certain series of compounds, the nature and size of their band gaps were not solely dependent on the crystal structure and composition of the elements. In more detail, the electron-precise Zintl-phase *A_5_Tt*P_3_ or *A*_10_*Tt*_2_P_6_ system (*A* = Li − Cs; *Tt* = Si − Sn) occurs with two different isomers for the anionic subunits—either carbonate analogue *Tt*P_3_^5−^ monomers with a partial *Tt*-P double bond character or *Tt*_2_P_6_^10−^ dimers with exclusively single bonds, respectively, (Figure 1). We found for all compounds with monomeric units a higher dispersion within the valence and conduction bands, which is in line with a partial double bond formation between the atoms of the carbonate analogue monomer *Tt*P_3_^5−^ anion. We also found no direct influence of the nature of the *Tt* element on the band gap. As a general trend, we found that direct band gaps at Γ are favoured in compounds with only little or no alkali metal orbital contribution at the valence band maximum, which suggests that the influence of the alkali metal on the electronic structure is an important factor [11].

With this new insight, it becomes evident that the matter at hand needs to be further investigated for a more diverse system, where the variety of compounds present is greater and the influence of the alkali metal on the crystal and band structure can be further studied. We report here on our investigation of a class of compounds with a large number of representatives, namely the Zintl phases with the composition *A*_3_*TrPn*_2_ (*A* = Li − Cs, *Tr* = Al, Ga, In; *Pn* = P, As). They were chosen since there are 22 experimentally achievable compounds, within seven different structure types. Although many of the compounds within the system were characterised in the 1990s, their electronic structures were only scarcely investigated for selected compounds [12,13,14]. An overview of the crystal structures known up to date is shown in Table 1 and Figure 2. According to the Zintl–Klemm concept, the alkali metals transfer their valence electron to the anionic substructure of the triel and pnictogen, thus allowing various covalent one-, two- or three-dimensional structural motifs of edge- or corner-sharing *TrPn*_4_-tetrahedra, as well as a “zero”-dimensional triangular planar coordination of the triel atom for compounds of the heavier alkali metals.

Here, we report on the structural relationship of the 22 experimentally known compounds and their electronic properties by calculating their band structures, density of states (DOS) and crystal orbital Hamilton population (COHP). The influence of the crystal structure and elemental composition on the nature and size of the band gaps is analysed. Further on, eight crystal structures for compounds with new element combinations were predicted and their electronic structures investigated. An overview of the similarities and differences between these structures is given.

## 2. Results and Discussion

### 2.1. Structural Relationship of Compounds with Composition A_3_TrPn_2_ (A = Li − Cs, Tr = Al − In, Pn = P, As)

The Zintl phases *A*_3_*TrPn*_2_ (*A* = Li − Cs, *Tr* = Al, Ga, In, *Pn* = P, As) possess a unique variety of structures due to the covalent bonding of two *Pn* atoms by one *Tr* atom, resulting in the same charge of the polyanions. Depending on the alkali metal triel/pnictide element combination, the anionic substructures adopt different dimensions, ranging from isolated molecular anions (“zero”-dimensional) up to three-dimensional networks. Representatives of the different structure types are shown in Figure 1.

Compounds with the largest alkali metals such as Cs_3_AlP_2_, Cs_3_GaP_2_, Cs_3_AlAs_2_, Cs_3_GaAs_2_ and Rb_3_GaP_2_ form “discrete” molecular polyanions. Whereas all Cs compounds crystalise in space group *Pbca* (no. 61, Figure 1A), Rb_3_GaP_2_ shows a lower monoclinic symmetry with space group *P*2_1_/*c* (no. 14, Figure 1B). These structures will be referred to as structure types **A** and **B**, respectively [15,16,17,18,19]. Despite the different symmetry of the crystals, both structure types contain the same edge-sharing triangular planar [*Tr*_2_*Pn*_4_]^6−^ polyanion, made up by two (3b-*Tr*)^0^, two (2b-*Pn*)^−^ and two (1b-*Pn*)^2−^ atoms (nb = n-fold bonded). In order to achieve an electron octet for all atoms, resonance structures with a *Tr-Pn* double bond are required (Figure 2, **1b**), resulting in four (2b-*Pn*)^−^ and two (4b-*Tr*)^1−^. Similarly, *Tt*P_3_ units have been observed in the *A*_5_*Tt*P_3_ (*A* = Li − Cs, *Tt* = Si − Sn) system, where [CO_3_]^2−^ isosteric [*Tt*P_3_]^5−^ units with a double bond character occur due to the higher amount of alkali metal [11]. The main difference between Rb_3_GaP_2_ and the Cs compounds is that the triangular dimers in Rb_3_GaP_2_ show a slightly bent structure, with a dihedral angle of 14.6(1)° between the two edge-sharing triangular planar *TrPn*_3_, compared to a perfectly planar arrangement for the Cs compounds.

K_3_AlP_2_, Rb_3_InP_2_ and K_3_AlAs_2_ form a structure type with two different polyanions and crystalise in space group *P*$\bar{1}$ (no. 2). They contain the molecular, triangular planar (“zero-dimensional”) [Al_2_*Pn*_4_]^6−^/[In_2_P_4_]^6−^ unit (Figure 2, **1a** and **1b**) that has been described above, and a linear one-dimensional substructure with tetrahedrally coordinated *Tr* atoms (Figure 2, **3**). The InP_4_ tetrahedra are edge-sharing, forming one-dimensional chains along the b-axis (Figure 1C), referred to as structure type **C** [20,21,22]. The chains contain exclusively (2b-Pn)^−^ and (4b-Tr)^−^, and the recurring unit can thus be formulated as [TrPn4/2]3−∞1. The two subunits are separated from each other through alkali metal atoms. Cs_3_InP_2_ shows in general the same structural motifs and crystallises in the same space group, but the one-dimensional chains and triangular planar units have a slightly different arrangement in the unit cell (Figure 1D) [23]. Here, the dimeric units of [In_2_P_4_]^6−^ have a planar structure, whilst the dihedral angles of 3.60(8)°, 4.0(2)° and 6.60(5)° in K_3_AlP_2_, Rb_3_InP_2_ and K_3_AlAs_2_, respectively, lead to a slightly bent conformation. This type is henceforth referred to as structure **D**.

The Na compounds, Na_3_AlP_2_ and Na_3_GaP_2_, Na_3_AlAs_2_, as well as K_3_InP_2_ and K_3_InAs_2_, are all isostructural and crystallise in space group *Ibam* (no. 72) [24,25,26,27,28]. They consist solely of one-dimensional chains of edge sharing tetrahedra [TrPn4/2]3−∞1 along the c-axis (**3**, SiS_2_-type analogue) (Figure 1E). The structure is henceforth referred to as structure type **E** [29].

Li_3_AlP_2_, Li_3_GaP_2_, Li_3_AlAs_2_ and Li_3_GaAs_2_ crystallise in space group *Cmca* (no. 64) and consist exclusively of dimers of edge-sharing *TrPn*_4_ tetrahedra. The *Tr*_2_*Pn*_6_ dimers (Figure 2, **2**) are further connected by their vertices to neighbouring units of the same type, with the *Pn*-*Pn* vector of each edge-sharing tetrahedra dimer alternating in perpendicular directions with respect to each other, thus forming two-dimensional layers (Figure 1F), henceforth referred to as structure type **F** [12,30,31]. Consequently, the two-dimensional polyanion [TrPn4/2]3−∞2 exclusively consists of (4b-*Tr*)^−^ and (2b-*Pn*)^−^, resulting in the same bonding situation as structure type **E**.

Li_3_InP_2_ and Li_3_InAs_2_, in contrast, form a three-dimensional structure with exclusively vertex-sharing *TrPn*_4_ tetrahedra in space group *I*4_1_/*acd* (no. 142) [31,32]. Four *TrPn*_4_ tetrahedra form an adamantane-type [In_4_Pn_10_] “super tetrahedra” (Figure 2). These units are further connected via their “outer vertices” to neighbouring super-tetrahedra to build a three-dimensional polyanion [TrPn4/2]3−∞3. The structure can be understood as a hierarchical variant of the diamond structure by replacing C atoms with adamantane-type units. Since each *Tr* and *Pn* atom remains four- and two-fold bonded, the same charge as in structures **E** and **F** results. Two such independent three-dimensional networks of tetrahedra (Figure 1G), referred to as structure type **G**, interpenetrate each other without any bonds formed between the two subunits.

Lastly, for Na_3_InP_2_ and Na_3_InAs_2_ an alternative three-dimensional connection of the InP_4_ tetrahedra is present [33,34]. These compounds crystallise in space group *P*2_1_/*c* (no. 14). Here In_2_*Pn*_6_ dimers of edge-sharing tetrahedra and In*Pn*_4_ tetrahedra (Figure 2, **2** and **4**, respectively) are connected via common *Pn* atoms. Since each *Pn*_4_*Tr*-tetrahedron is either corner- or edge-sharing with its neighbours, all *Pn* atoms are (2b-*Pn*)^−^, while In atoms remain four-fold bonded (4b-In)^−^, and the same overall charge of the polyanion [TrPn4/2]3−∞3 results (Figure 1H, referred to as structure type **H**).

Thus, all structure types presented can be considered as constitutional isomers of compounds with the composition *A*_3_*TrPn*_2_, since the bonding situation of the *Pn* and *Tr* atoms is the same, but their connection leads to a variety of zero-, one-, two-, and three-dimensional polyanions.

### 2.2. Crystal Structure Optimisation of the Known Compounds

Using the experimental crystal structures as starting models, all existing compounds of the *A*_3_*TrPn*_2_ system were structurally optimised with subsequent frequency calculations. All but one experimentally observed structure were proven to be true local minima. For Cs_3_InP_2_, one imaginary frequency at −12.7551 cm^−1^ was found, and thus, upon distortion along that frequency, a new optimisation in P 1 symmetry was conducted. This structure was then proven to be a true local minimum by the absence of further imaginary frequencies. However, the reduction in the symmetry did not lead to significant structural differences and did not change the band structure or density of states (DOS) at all.

Table 1 shows an overview of all experimental and calculated cell parameters for all existing compounds. They are in good agreement, with deviations below 2%, but for structure type **E**, the calculated cell parameter b can be up to 6.5% smaller than the experimental reference. This could be due to the anisotropy of the structure, with the one-dimensional chains being aligned along the c-axis. Thus, only the alkali metal “connects” them along b. Since the interactions between the chains are only weak, this parameter might shrink more compared to the other cell parameters, for the structure optimisation at 0 K.

### 2.3. Structure Predictions for Unknown A_3_TrPn_2_ Phosphides and Arsenides

The crystal structures of K_3_GaP_2_ Rb_3_AlP_2_, Na_3_GaAs_2_, K_3_GaAs_2_, Rb_3_AlAs_2_, Rb_3_GaAs_2_, Rb_3_InAs_2_ and Cs_3_InAs_2_ have not yet been reported. In order to make a theoretical prediction of the stability of the possible structure types, models were constructed based on the crystal structures of experimentally feasible P and As compounds as described above, containing either alkali, triel or pnictide atoms of the previous or next period. For example, for K_3_GaP_2_, models based on Rb_3_GaP_2_, K_3_AlP_2_ and Na_3_GaP_2_ with structure types **B**, **C** and **E**, respectively, were used, since these are the experimentally feasible compounds with the lighter or heavier alkali metal (Na_3_GaP_2_ and Rb_3_GaP_2_, respectively) or triel element (K_3_AlP_2_). From their experimental crystal structures, models were created for K_3_GaP_2_, optimised and the subsequent frequency, band structure, density of states (DOS) and crystal orbital Hamilton population (COHP) calculated. After optimisation and frequency calculation, the energy per unit cell and Gibbs free enthalpy can be obtained, respectively, and compared between each compound. Results for all structure predictions can be found in Table 2 and Appendix A, where the differences (ΔΔE and ΔΔG) were referenced to the lowest energy ΔE and Gibbs free enthalpy ΔG for each compound.

Thus, the predicted structures are as follows: For K_3_GaP_2_, R_3_AlP_2_, K_3_GaAs_2_, Rb_3_GaAs_2_ and Cs_3_InAs_2_, structure type **B** with its trigonal planar dimer units has the lowest energy and Gibbs free enthalpy and is therefore considered to be the most stable. The lowest energy for Na_3_GaAs_2_ corresponds to structure type **H**, with its three-dimensional network of corner and edge-sharing tetrahedra. For Rb_3_InAs_2_, two structure types are possible, since the lowest energy could be obtained for structure **C**, while the mixed zero- and one-dimensional structure and the lowest Gibbs free enthalpy could be obtained for structure type **A**. This could be an indication of a possible phase transition from structure **C** to **A** for an increased reaction temperature and subsequent quenching.

Since all the computed energy and enthalpy differences are small, phase transitions between these predicted structures might occur. For example, the difference for Rb_3_AlP_2_ between the favoured type **B** and type **A** is only 6.21 kJ/mol. Although higher than the difference to structure **C**, this difference decreases for the Gibbs free enthalpy, which—in contrast to the bare minimum energy calculated at 0 K—includes the influence of the temperature for the optimised structure. In the case of Rb_3_AlP_2_, the decrease in ΔΔG for structure type **A**, compared to the energy difference, hints towards the existence of a high-temperature phase. The same trend is found for type **E** Na_3_GaAs_2_, type **A** Rb_3_AlAs_2_ and type **A** Cs_3_InAs_2_, which therefore could also be candidates for high-temperature phase transitions.

The general trend in dimensionality of the anionic sublattice shows that larger alkali metal atoms prefer a lower dimensionality, but larger triel atoms (mostly In) prefer a higher dimensionality. These trends can be seen, for example, in the Li-Al and Li-Ga phosphides and antimonides, which all crystallise in the two-dimensional structure **F**, while their corresponding In compounds crystallise in the three-dimensional structures **G** and **H**. Furthermore, all Rb and Cs compounds crystallise in structures containing only edge-sharing triangular planar dimers, type **A** and **B**, except Cs_3_InP_2_, Rb_3_InP_2_ and Rb_3_InAs_2_ (based on energy comparison), which crystallise in structures **C** or **D**, containing additional one-dimensional substructures. The remaining Na and K compounds crystallise in the one-dimensional or one- and zero-dimensional structure types **E** and **C**, except for the predicted structures of K_3_GaP_2_, K_3_GaAs_2_ and Na_3_GaAs_2_ which prefer structure types **B** and **H**, respectively. Comparing the structures of each phosphide–antimonide pair, almost all show the same structure, except for Na_3_GaP_2_/Na_3_GaAs_2_ and Cs_3_InP_2_/Cs_2_InAs_2_.

From these trends, it seems clear that the size of the atoms is the main factor determining which crystal structure is formed. More complex structures, formed through connections of *TrPn*_4_-tetrahedra, allow only for smaller voids, where Li and Na ions fit best. The larger cations K, Rb and Cs are more space demanding, and may not fit in these voids. Instead, less complex anionic substructures with a more flexible anion arrangement are formed (such as monomeric units) into which the larger cations fit better. The second trend of the more complex structures for In compounds, can also be explained by the larger size of the In atoms. Larger atoms widen the polyanionic network, resulting in larger voids for the cations. Al and Ga cannot widen the structure enough; thus, the simpler polyanionic structures **A** and **B** are formed for the Rb and Cs compounds. 

### 2.4. Electronic Structure Analysis

#### 2.4.1. Size of the Band Gap

For all compounds and predicted structures, the band structure and the density of states (DOS) were calculated. Plots for all structures can be found in the SI, and all band gap sizes and band gap types are listed in Table 3. If an indirect band gap is less than 0.03 eV smaller than the smallest direct band gap, the gap is denoted as a pseudo-direct band gap. As an interesting fact, we found that 27 compounds reveal a direct or pseudo-direct band gap and only Na_3_AlP_2_, Na_3_GaP_2_ and Na_3_AlAs_2_ possess indirect band gaps.

The width of the band gaps is decreasing for all heavier triels, including the most stable predicted structures of the eight unknown compounds, except for K_3_AlP_2_ and K_3_AlAs_2_, which have much smaller band gaps than their heavier homologues with Ga and In. Since these structures possess [Al_2_P_4_]^6−^ dimeric units, which incorporate partial double bonds (Figure 2), in contrast to structures with solely tetrahedral structural motifs, smaller band gaps are expected. This is indeed found for all structures incorporating this motif, including the predicted compounds, which also show especially small band gaps for compounds with structure type **C**. For Cs_3_GaAs_2_, the electronegativity difference could explain the slight increase in the band gap, since Ga has a higher electronegativity [35].

As for the comparison between the band gaps of phosphides and arsenides, the latter have, in general, gaps that are about 0.2 to 0.4 eV smaller than the corresponding phosphides. This observation correlates with the idea that for the larger As atoms, the orbital overlap with their neighbouring atoms is less effective, which leads in molecules to a smaller HOMO-LUMO separation. However, three compounds do not follow this trend, namely Na_3_Ga*Pn*_2_, for which the As compound shows a much smaller band gap, as well as Cs_3_In*Pn*_2_ and Rb_3_In*Pn*_2_ (lower enthalpy structure), for which the As compounds have a larger band gap. Interestingly, this coincides with the set of compounds for which the structure type of the phosphide and arsenide are different, which raises the question as to whether the structure type also has an influence on the size of the band gap. A closer look at the difference between the calculated bad gaps for the predicted structures confirms this hypothesis, since the band gaps for the different structure types of, for example, Na_3_GaAs_2_ range from 2.03 eV (**H**) to 2.39 eV (**E**).

By taking a closer look at the band gap size distribution over the different dimensionalities of structure types **A** to **H**, the band gap is on one hand increasing with the dimension of the anionic structure and on the other decreasing with the occurrence of edge-sharing tetrahedra as an anionic structural motif. Therefore, the lowest band gaps are obtained for structure types **C** and **D**, with a zero- and one-dimensional structure, followed by types **A** and **B** (pure zero-dimensional triangular planar units), type **E** (only one-dimensional), types **F** and **H** (two- and three-dimensional) and lastly type **G** (three-dimensional structure without edge-sharing tetrahedra). This could explain the larger band gaps for the As compound of Cs_3_In*Pn*_2_ and Rb_3_In*Pn*_2_, since they crystallise with molecular anions (zero-dimensional structures). If they were to adopt the same structure type as their corresponding P compounds, the band gaps of the As compounds would follow the trend of having lower gaps. 

All band structures and DOS plots are available in the SI. Among the 30 compounds, we chose 4 examples to show the characteristics of the various band structures. Li_3_AlP_2_ and Cs_3_AlP_2_, with structure type **F** and **A**, respectively, shown in Figure 2, correspond to compounds with direct and pseudo-direct band gaps. Na_3_AlP_2_ and K_3_InlP_2_, shown in Figure 3, are an example where both structures adapt the same structure type **E** but have indirect and direct band gaps, respectively. 

The dispersion of the bands varies for the different structure types **A** and **F**. As for type **F**, all structure types with two- or three-dimensional polyanions (types **E**–**H**, Figure 2a) show a large band dispersion (especially in the area of the filled band region below E_F_), whereas structures that incorporate “discrete” edge-sharing triangular planar dimers (types **A**–**D**, Figure 2b) have bands with a small dispersion (flat bands). As a consequence, all Cs compounds show an increased occurrence of pseudo-direct band gaps as they adopt structure types **A**, **C**, and **D**. Since especially the top valence band is very flat at the Fermi level, multiple transitions similar in energy are possible and pseudo-direct band gaps arise. As an example, the band structure of Cs_3_AlP_2_, with two possible transitions, is shown in Figure 2b. The direct band gap at Γ (green transition) is only 0.02 eV larger than the indirect transition Y_2_ → Γ (orange arrow). The low dispersion of the bands is a result of the rather localised electron density within the anionic–molecular [*Tr*_2_*Pn*_4_]^6−^ units that are fully separated by alkali metal ions, while extended edge- or corner-sharing tetrahedra structural motifs result in a farther-distributed electron density along the two- or three-dimensional network, resulting in more disperse bands. 

#### 2.4.2. Band Structures and DOS

The DOS looks similar for all *A*_3_*TrPn*_2_ compounds: the valence bands contain mostly *Pn* states with minor contributions of the alkali and triel elements. Around the Fermi level of compounds with flat bands, an increase in alkali metal states can be observed. The lower conduction bands for most compounds contain mostly states of the triel, followed by pnictide and alkali metal states. This is not the case for Na_3_AlP_2_, Na_3_GaP_2_, Na_3_AlAs_2_ and K_3_InP_2_, where all elements contribute equally to the DOS. For higher bands, the most contributing element either changes, or all elements contribute equally.

All direct and pseudo-direct band gaps are located at Γ. Only three compounds, Na_3_AlP_2_ (Figure 3a), Na_3_GaP_2_ (Appendix A, SI) and Na_3_AlAs_2_ (Appendix A, SI), have an indirect band gap. For Na_3_AlP_2_, the valence band minimum (VBM) appears—in contrast to, for example, K_3_InP_2_ (Figure 3b)—not at Γ but on the path from Γ to X. At the same time, the lowest conduction band gets pushed up at Γ such that the conduction band minimum (CBM) lies between G_0_ and X (Figure 3a). Na_3_GaP_2_ again has the VBM at Γ while the CBM remains at the same position (Appendix A). A closer look at the DOS shows that, in contrast to all direct band gap compounds at the Fermi level, additional alkali metal states are present in indirect band gap compounds, leading to changes in the VBM. An increase in alkali metal states is also observed for compounds with very flat valence bands at the Fermi level. 

The same can be seen for the conduction bands, which are pushed up in Na_3_AlP_2_, Na_3_GaP_2_ and Na_3_AlAs_2_ at Γ. In the DOS, it can be seen that there are more Na states present. The Li compounds and Na_3_InP_2_/Na_3_InAs_2_ have mainly triel states, followed by small contributions of Li/Na and P/As. Compounds with the heavier alkali metals also have more alkali metal states around the CBM, although they are at or close to Γ, so the mere existence of these states should not prevent the occurrence of direct gaps. This suggests that the interaction of the alkali metal with itself or adjacent triel and pnictide atoms could be a determining factor for the occurrence of direct and indirect band gaps.

A Mulliken analysis was conducted for all compounds, as well as all predicted compounds, determining the atom’s Mulliken charges and the overlap population between neighbouring atoms (see SI). Mulliken charges represent trends in the charge distribution based on the calculated wave function, but do not reproduce the full expected ionic charges or formal charges that occur in representations of Lewis formulae. In all compounds with two-fold bonded *Pn* atoms, P and As atoms have a Mulliken charge of about −1. For those compounds that contain both one- and two-fold bonded Pn atoms (1b and 2b, respectively), the 1b-Pn have up to 0.2 higher negative charges, thus reflecting the trend of the bonding situation as shown in Figure 2. All alkali metals show Mulliken charges, which are increasing for Na and K from about 0.56 to 0.75, but decrease again for Rb and Cs to around 0.70. Overall, the expected positive charge of the alkali metal is reflected in those numbers. 

With respect to chemical bonding, triel elements can on the one hand be treated as *Tr*^3+^ according to an ionic description and, on the other hand, with a formal charge of −1 or 0, according to a covalent description applying the octet rule, for the tetrahedral and triangular planar coordination, respectively. The Mulliken charges calculated for the triel atoms are between −0.3 and +0.5. The values represent a mostly polar covalent description of the charge, and the Mulliken charge of the triel element decreases for compounds with smaller electronegativity differences. Therefore, Al compounds show the highest charges. On the other hand, a trend for between three- and four-fold-bonded structure motifs can be found for each triel atom, since the atoms which are tetrahedrally coordinated show larger Mulliken charges than the triangular planar ones, due to the higher number of polar bonds formed.

Values larger than 0.2 for the overlap population indicate bonding triel–atom *Pn* interactions, which are in line with the bonds assigned based on the interatomic distances described in the crystal structures. Further on, the highest overlap populations are found for *Tr*-*Pn* bonds, which form partial double bonds within the dimeric triangular planar units in structures **A** to **D** (see Figure 2). Here, for example, the Al-P overlap in K_3_AlP_2_ is about 0.5, which strongly hints towards a higher bond order. Since the absolute value of the overlap population for these partial double bonds decreases for Ga-*Pn* and In-*Pn*, the validation of the double bond rule, namely that heavier elements are less favoured for double bond formation, is stated. Since the probability of forming (partial) double bonds decreases with the atomic number within a group, Ga and In show less overlap population for these atomic interactions [36].

Between alkali metals and *Pn*, as well as alkali metals and triel, only negligible interactions are found. For the interactions between two neighbouring triel elements, negative values were obtained for the overlap population, which suggests that atoms in adjacent (edge-sharing) tetrahedra and triangular planar units have no attractive, but a slightly repelling, interaction. This is more distinct for compounds of the heavier triel atoms, due to their increase in size, as well as structures with edge-sharing tetrahedra. As a consequence, the distance between adjacent triel atoms is smaller. The same negative overlap, and thus repulsion, can be found for the long-distance interaction between the next but one *Pn* atoms.

For all compounds, a crystal orbital Hamilton population (COHP) was calculated for all heteroatomic interactions, as well as the triel–triel interaction(s). For the latter, as well as the extra plots of the *Tr*-*Pn* interactions, they were reduced to the interactions of only neighbouring (and thus, for *Tr*-*Pn*, between bonding) atoms. A selection of these COHP plots can be found in Figure 4. In general, as seen in Figure 4a, there are only few interactions from 0 eV until about −3 eV in the COHP, although a lot of states are present in the DOS, especially for *Pn*. Therefore, in this energy range, mainly the states of Pn lone pairs contribute to the DOS. 

For Na_3_Al*Pn*_2_ and Na_3_GaP_2_ the projected COHP of the heteroatomic interactions shows two peculiar features. Below the Fermi level, there occur, compared to the other compounds, low values for the Na-*Tr* and Na-*Pn* interactions. The higher number of states observe in the DOS hints at additional Na electron density close to the Fermi level, which was deemed to be responsible for the position of the valence band maximum (VBM) and thus the indirect band gap, as pointed out above. Since the interactions with Na are small, the additional states are thus attributed to non-bonding Na states (see SI). Secondly, the COHP of the Al-Al interactions (Figure 4b, left) shows a sharp increase in states at the band gap top. These interactions between the triel atoms might be responsible for the position of the conduction band minimum (CBM) at another *k*-point than Γ, since they are not present in K_3_InP_2_ (Figure 4b, right), which also crystalises in the same structure type, but has a direct band gap (at Γ). The stronger In-In interactions, which are also represented by a stronger negative overlap population in the Mulliken analysis, could lead to an increase in energy for these interactions and thus an absence of them above the band gap. 

Comparing the COHP of *Tr-Tr* interactions of structure type **E** with all other compounds, interactions between the triel atoms are absent or have a similar shape as K_3_InP_2_, for which they change between bonding and anti-bonding interactions above the Fermi level. Since all these compounds also show direct band gaps, the strength of the *Tr-Tr* interaction might be able to influence the position of the CBM. To gain more insight into this hypothesis, calculations with post-DFT methods have to be conducted, since the level of theory present is limited in the accuracy of describing excited states. Further on, relativistic effects might play a role, since within structure type **E** the switch to direct band gaps occurs only for In compounds. Since the In basis set accounts for some relativistic effects by a core potential, which the Al and Ga basis sets do not, the direct band gap could also arise from the chosen level of theory. 

Lastly, all compounds that contain [*Tr*_2_*Pn*_4_]^6−^ units show another interesting feature in the “bond” projected COHP of *Tr*-*Pn* interactions. Figure 4c (left) shows that for Li_3_AlP_2_, the COHP, calculated for interactions of Al-P bonds, no significant difference between the different bonds can be found, since all Al-P bonds are single bonds. Therefore, the shape of the projected COHPs is the same. For Cs_3_AlP_2_ (Figure 4c, right), due to the partial double bond for the terminal P atoms P1 and P2, their interactions can be found directly below the Fermi level, since these π-bonds are high in energy. The bonding interactions for the bridging P atoms, P3 and P4, are lower in energy (at about −2 eV) due to the single bonds. 

## 3. Methods

The computational studies of all compounds in the *A*_3_*Tr*P_3_ system (with *A* = Li − Cs and *Tr* = Al, Ga, In) were performed using the CRYSTAL17 program package and hybrid density functional methods [37,38]. A hybrid exchange–correlation functional after Perdew, Burke, and Ernzerhof (DFT-PBE0) was used, ref. [39] Localised, Gaussian-type triple ζ-valence + polarisation level basis sets were used for Al, Ga, In and P and split valence + polarisation level basis sets for Li, Na, K, Rb and Cs. The basis sets were derived from the molecular Karlsruhe basis sets [12,28,40,41,42]. For the evaluation of the Coulomb and exchange integrals (TOLINTEG), tight tolerance factors of 8, 8, 8, 8, 16 were used for all calculations. The reciprocal space of all calculations was sampled with Monkhorst–Pack-type *k*-point grids; their respective sizes can be found in the SI. The starting geometries were taken from experimental data whenever possible. For the unknown compounds, models based on adjacent structures by atom replacement were derived. Both the lattice parameters and atomic positions were fully optimised within the constraints imposed by the space group symmetry. Further on, all optimised structures were confirmed to be true local minima by means of harmonic frequency calculations at the Γ-point. For all compounds and models, electronic band structures and density of states (DOS) were calculated. The Brillouin Zone paths were provided by the web service *SeeK-path* and a list can be found in the SI [43].

## 4. Conclusions

In this paper, band structures and density of states for all 30 *A*_3_*TrPn*_2_ compounds with *A* = Li, Na, K, Rb, Cs; *Tr* = Al, Ga, In; *Pn* = P, As, a family of electron-precise Zintl phases, were presented. A total of 27 out of 30 compounds show direct or pseudo-direct band gaps. Na_3_AlP_2_, Na_3_AlAs_2_ and Na_3_GaP_2_ possess an indirect band gap. For the compounds with indirect band gaps, the conduction band minimum (CBM) and valence band maximum (VBM) are not located at Γ (as for the other compounds). The position of the VBM is probably caused by an increase in (non-bonding) Na states right below the Fermi level. The position of the CBM could be influenced by the strength of the *Tr*-*Tr* interaction above the band gap as well as relativistic effects, which are partially accounted for in the calculation by the In basis set. The width of the band gap was discussed, based on the structure as well the composition of the compound. Arsenides, as well as compounds with lower-dimensional anionic substructures, show lower band gaps, due to less interatomic overlap within the structure or between the atoms.

The bonding situation imposed by the crystal structure was confirmed by a Mulliken analysis, where bonds between the triel and *Pn* atoms were found. For the triangular planar *Tr*_2_*Pn*_4_ building units, partial double bonds for the lighter triels were identified, as well as being excluded for the compounds with the heavier element In according to the double bond rule. Insights into atomic interactions, by calculation of the COHP, further described the electronic situation of the compounds. 

For confirmation as well as to expand the correlations found, experiments on synthesising the predicted compounds need to be conducted. Experimentally known compounds should also be screened for possible phase transitions to gain an even deeper insight into the system. Calculations and structure predictions could also to be extended to antimonides and bismuthides to see how they affect the appearance of direct band gaps, since both basis sets take relativistic effects into account with a core potential [44,45,46,47]. Calculations should also be expanded to compounds in the *A*_2_*TtPn*_2_ system (with *A* = Li−Cs, *Tt* = Si−Sn, *Pn* = P, As), which is the equivalent tetrel system, and therefore their crystal structures are closely related [48,49,50,51,52].

Further on, post-DFT calculations are needed to investigate all the effects found for conduction bands, as they provide a more accurate description for excited states [53,54].

## Data Availability

Data are contained within the article and Appendix A.

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
