# Peer review of "Large Number of Direct or Pseudo-Direct Band Gap Semiconductors among A3TrPn2 Compounds with A = Li, Na, K, Rb, Cs; Tr = Al, Ga, In; Pn = P, As"

_molecules, 2024, doi:10.3390/molecules29174087_

Round 1

Reviewer 1 Report

Comments and Suggestions for Authors

In this manuscript “Large number of direct or pseudo-direct band gap semiconductors among A3TrPn2 compounds with A=Li, Na, K, Rb, Cs; Tr = Al, Ga, In; Pn=P, As”, the authors performed first-principle calculations to study 22 combinations of A3TrPn2. They found several zero-, one-, two-, and three-dimensional polyanionic substructures (A-H) and then analyzed their electronic properties such as electronic band structures, density of states, and COHP. This study provides insight for the design of new semiconductors. Overall, this manuscript is suitable for publication in Molecules with minor revisions:

1)     The structure coordinates information should be given for each type of structure (at least in SI), not just the volume information

2)     Is there any particular reason why type A structure is listed as the last and some of the structure types were disordered? (in Figure A and in SI) 

Author Response

We like to thank all reviewers for their overall positive statements and their valuable comments. All significant changes are highlighted with yellow background. Typos and grammar corrections are included as well.

Comments and Suggestions for Authors

In this manuscript “Large number of direct or pseudo-direct band gap semiconductors among A3TrPn2 compounds with A=Li, Na, K, Rb, Cs; Tr = Al, Ga, In; Pn=P, As”, the authors performed first-principle calculations to study 22 combinations of A3TrPn2. They found several zero-, one-, two-, and three-dimensional polyanionic substructures (A-H) and then analyzed their electronic properties such as electronic band structures, density of states, and COHP. This study provides insight for the design of new semiconductors. Overall, this manuscript is suitable for publication in Molecules with minor revisions:

1)     The structure coordinates information should be given for each type of structure (at least in SI), not just the volume information

Answer: We added a table with all atomic coordinates in the SI.

2)     Is there any particular reason why type A structure is listed as the last and some of the structure types were disordered? (in Figure A and in SI) 

Answer: I suppose this question refers to Table 1 (since there is no Figure A). Figure 1 is ordered according to the structure descriptions in the introduction, while all tables list the structure types from lighter atoms (Li) to heavier atoms (Cs). It is not clear to us what is the meaning of “structure types were disordered”, since only for Cs3InP2 a disordered model was calculated due to the imaginary frequency.

Reviewer 2 Report

Comments and Suggestions for Authors

The authors present a detailed and extensive investigation of the compound family A3TrPn2 by band structure methods. The study is well organized and systematic. The manuscript is clearly structured and well-written. Figures and tables are well-selected and clear, title and abstract are concise and not misleading. The supporting material is comprehensive with numerous figures and tables comprising all relevant information, nevertheless well organized and accessible. 

Thus, I have only a few minor remarks for having to be considered by the authors:

Abstract

Manuscript: Due to the high impact of semiconductors with respect to many applications for electronics and energy transformation the search for new compounds and a deep understanding of the structure-property relationship in such materials has a high priority.

Suggested change (add a comma after transformation): Due to the high impact of semiconductors with respect to many applications for electronics and energy transformation,  the search for new compounds and a deep understanding of the structure-property relationship in such materials has a high priority.

2.3.

Manuscript: The general trend in dimensionality of the anionic sublattice shows is that ...

Suggested change (add a plural s and delete the "is"): The general trend in dimensionality of the anionic sublattices shows that

2.4.1.

Manuscript: For all compounds as well as the predicted structures band structure and the density of states (DOS) were calculated.

Suggested change: For all compounds and predicted crystal structures, band structures and the corresponding the density of states (DOS) were calculated.

2.4.2. (in the part of the Mulliken analysis)

Manuscript: The Mulliken charges calculated for the triel atoms are between - 0.3 and + 0.5. The values represent a mostly ionic description of the charge, ...

Question and comment: I am not sure whether I understand the drawn conclusion correctly or not. Anyway, I would prefer to conclude that charges of the triel element covering the range from -0.3 to +0.5 (in combination with significant overlap populations Tr-Pn) rather support  a covalent (or polar covalent) picture of the bonding than an ionic picture. The statement "represent a mostly ionic description" is the only conclusion of the paper which I can't share. It is not that I simply insist on my point of view; I rather do not see the reasons why the author prefer the ionic picture.

4. Conclusions

In the final paragraph, the word "further" (once misspelled as furter) is repeated four times. Please rephrase.

In conclusion, I recommend to accept the manuscript as a feature paper in molecules after the authors have considered the remarks given above.  

Author Response

We like to thank all reviewers for their overall positive statements and their valuable comments. All significant changes are highlighted with yellow background. Typos and grammar corrections are included as well.

Comments and Suggestions for Authors

The authors present a detailed and extensive investigation of the compound family A3TrPn2 by band structure methods. The study is well organized and systematic. The manuscript is clearly structured and well-written. Figures and tables are well-selected and clear, title and abstract are concise and not misleading. The supporting material is comprehensive with numerous figures and tables comprising all relevant information, nevertheless well organized and accessible. 

Thus, I have only a few minor remarks for having to be considered by the authors:

Abstract

Manuscript: Due to the high impact of semiconductors with respect to many applications for electronics and energy transformation the search for new compounds and a deep understanding of the structure-property relationship in such materials has a high priority.

Suggested change (add a comma after transformation): Due to the high impact of semiconductors with respect to many applications for electronics and energy transformation,  the search for new compounds and a deep understanding of the structure-property relationship in such materials has a high priority.

Answer: Corrected

2.3.

Manuscript: The general trend in dimensionality of the anionic sublattice shows is that ...

Suggested change (add a plural s and delete the "is"): The general trend in dimensionality of the anionic sublattices shows that

Answer: Corrected

2.4.1.

Manuscript: For all compounds as well as the predicted structures band structure and the density of states (DOS) were calculated.

Suggested change: For all compounds and predicted crystal structures, band structures and the corresponding the density of states (DOS) were calculated.

Answer: Corrected

2.4.2. (in the part of the Mulliken analysis)

Manuscript: The Mulliken charges calculated for the triel atoms are between - 0.3 and + 0.5. The values represent a mostly ionic description of the charge, ...

Question and comment: I am not sure whether I understand the drawn conclusion correctly or not. Anyway, I would prefer to conclude that charges of the triel element covering the range from -0.3 to +0.5 (in combination with significant overlap populations Tr-Pn) rather support  a covalent (or polar covalent) picture of the bonding than an ionic picture. The statement "represent a mostly ionic description" is the only conclusion of the paper which I can't share. It is not that I simply insist on my point of view; I rather do not see the reasons why the author prefer the ionic picture.

Answer: We agree and it was intended to have a primarily covalent description, with trends representing the polarity of the bonds. We rephrased this part.

  1. Conclusions

In the final paragraph, the word "further" (once misspelled as furter) is repeated four times. Please rephrase.

Answer: Thank you for pointing this out. We rephrased that part.

In conclusion, I recommend to accept the manuscript as a feature paper in molecules after the authors have considered the remarks given above.  

Reviewer 3 Report

Comments and Suggestions for Authors

The authors performed all necessary DFT calculations to optimize the structures of 30 electron precise Zintl compounds, which include prediction of crystal structures of three compounds. The electronic band structure, density of states, and crystal orbital Hamilton populations were also looked into.  Their considerable efforts are appreciated.  Meanwhile, the work can be  improved.

(1) section 2.1 can't be regarded as results. It should be moved to the introduction part.

(2) kJ/mol should be changed to eV, since eV is more common in Computational  Chemistry, and the unsubstantial difference between close structures can be more distinct.

(3) ∆∆G (enthalpy) in Table 2 is not necessary since computed energy is used to determine the most stable structure. No further discussion on the enthalpy was given in the manuscript.

(4) Table 1 lists 30 compounds, while Table S1 only lists 21 compounds.  And more, the K-point density for different structures show substantial difference. 

(5) The relation between atomic size to crystal structure or band structure is qualitatively described. A quantitative relation between them will provide more valuable information.

Comments on the Quality of English Language

There are some typos and many grammar mistakes in the manuscript. For example, line 27 page 1, "divers" and the sentence in line 33 page 1. 

The conclusions part is not well expressed. For example, the authors should give their findings on the bonding situation directly in line 479 page 17.

Author Response

We like to thank all reviewers for their overall positive statements and their valuable comments. All significant changes are highlighted with yellow background. Typos and grammar corrections are included as well.

Comments and Suggestions for Authors

The authors performed all necessary DFT calculations to optimize the structures of 30 electron precise Zintl compounds, which include prediction of crystal structures of three compounds. The electronic band structure, density of states, and crystal orbital Hamilton populations were also looked into.  Their considerable efforts are appreciated.  Meanwhile, the work can be  improved.

(1) section 2.1 can't be regarded as results. It should be moved to the introduction part.

Answer: Indeed, we describe here the structural relationship between the 22 observed structures which was not comprehensively done before. We like to keep this in the result section, since we follow consequently in the here described groups of structures. We changed the title of the section accordingly.

(2) kJ/mol should be changed to eV, since eV is more common in Computational  Chemistry, and the unsubstantial difference between close structures can be more distinct.

Answer: The usage of kJ/mol or eV is both commonly used. We like to stay with kJ/mol, since units can be easily be converted by the readers.

(3) ∆∆G (enthalpy) in Table 2 is not necessary since computed energy is used to determine the most stable structure. No further discussion on the enthalpy was given in the manuscript.

Answer: We used the ΔΔG values to discuss possible phase transitions at elevated temperatures. We have pointed this out more clearly and rephrased two sentences.

(4) Table 1 lists 30 compounds, while Table S1 only lists 21 compounds.  And more, the K-point density for different structures show substantial difference. 

Answer: We added the missing k-paths to table S1 (now S2).

The k-point density we used here is determined by the Crystal17 program package, which in general uses a well-accepted spacing between different k-points. The density correlates with the length of the unit cell  (or corresponding length of reciprocal cell).

(5) The relation between atomic size to crystal structure or band structure is qualitatively described. A quantitative relation between them will provide more valuable information.

Answer: This is a great idea, which we are following on a long-term, besides other correlations. We found, that we need more data to benchmark the limits of interpretation.We are preparing other studies of related systems and we are constantly traying to correlate several parameters. This paper already covers a lot of data. Even though we describe many structures here, we like to put a quantitative analysis on a larger number and this currently goes beyond the scope of this article.

Comments on the Quality of English Language

There are some typos and many grammar mistakes in the manuscript. For example, line 27 page 1, "divers" and the sentence in line 33 page 1. 

The conclusions part is not well expressed. For example, the authors should give their findings on the bonding situation directly in line 479 page 17.

Answer: We corrected the paper for grammar mistakes and typos and rephrased the Conclusion.